# The Role of MacroH2A Histone Variants in Cancer

**DOI:** 10.3390/cancers13123003

**Published:** 2021-06-15

**Authors:** Chen-Jen Hsu, Oliver Meers, Marcus Buschbeck, Florian H. Heidel

**Affiliations:** 1Internal Medicine C, Greifswald University Medicine, 17475 Greifswald, Germany; chen-jen.hsu@med.uni-greifswald.de; 2Cancer and Leukaemia Epigenetics and Biology Program, Josep Carreras Leukaemia Research Institute (IJC), Campus Can Ruti, 08916 Badalona, Spain; omeers@carrerasresearch.org; 3Program for Predictive and Personalized Medicine of Cancer, Germans Trias i Pujol Research Institute (PMPPC-IGTP), Campus Can Ruti, 08916 Badalona, Spain; 4Leibniz Institute on Aging, Fritz-Lipmann Institute, 07745 Jena, Germany

**Keywords:** macroH2A, histone variants, epigenetics, chromatin, cancer, macrodomain, tumor suppressor, oncohistone, malignant transformation

## Abstract

**Simple Summary:**

The structural unit of chromatin is the nucleosome that is composed of DNA wrapped around a core of eight histone proteins. Histone variants can replace ‘standard’ histones at specific sites of the genome. Thus, histone variants modulate all functions in the context of chromatin, such as gene expression. Here, we provide a concise review on a group of histone variants termed macroH2A. They contain two additional domains that contribute to their increased size. We discuss how these domains mediate molecular functions in normal cells and the role of macroH2As in gene expression and cancer.

**Abstract:**

The epigenome regulates gene expression and provides a molecular memory of cellular events. A growing body of evidence has highlighted the importance of epigenetic regulation in physiological tissue homeostasis and malignant transformation. Among epigenetic mechanisms, the replacement of replication-coupled histones with histone variants is the least understood. Due to differences in protein sequence and genomic distribution, histone variants contribute to the plasticity of the epigenome. Here, we focus on the family of macroH2A histone variants that are particular in having a tripartite structure consisting of a histone fold, an intrinsically disordered linker and a globular macrodomain. We discuss how these domains mediate different molecular functions related to chromatin architecture, transcription and DNA repair. Dysregulated expression of macroH2A histone variants has been observed in different subtypes of cancer and has variable prognostic impact, depending on cellular context and molecular background. We aim to provide a concise review regarding the context- and isoform-dependent contributions of macroH2A histone variants to cancer development and progression.

## 1. Introduction

Chromatin structure is the template for transcriptional regulation and chromatin-based modifications provide the molecular basis for an epigenetic memory affecting physiologic cellular functions such as proliferation, differentiation, and cell cycle [1]. In cells, only a small fraction of the 6 billion DNA bases comprising the genome is accessible to the transcriptional machinery. The remainder is compacted and sequestered by hierarchical folding of DNA into compacted chromatin. The configuration of chromatin, including nucleosome positioning and three-dimensional folding, has an impact on the accessibility of DNA for transcription factor binding [2]. The regulation of chromatin structure occurs on multiple levels and contributes to correct temporal and spatial transcriptional programs that are essential for cell identity in the organism. Disruption of chromatin homeostasis is a hallmark of cancer and often promotes oncogenic gene expression. Targeting of cancer-associated epigenetic changes has already been successful for therapeutic intervention. Epigenetic modulators called ‘epi-drugs’ (e.g., inhibitors of DNA methyltransferases and histone deacetylases) have been approved for the treatment of different cancers [3,4,5]. Furthermore, epigenetic marks may serve as biomarkers for diagnosis, prognosis and prediction of disease recurrence [6,7].

The structural unit of chromatin is the nucleosome. In eukaryotic cells, the nucleosome consists of approximately 147 base pairs of genomic DNA wrapped around an octameric core of histones H2A, H2B, H3 and H4 [8,9]. The nucleosome is further stabilized by the binding of linker histone H1 bound at the entry and exit sites of the DNA [10]. The N-terminal tails of all core histones and the C-terminal tail of H2A histones protrude out of the complex structure of the nucleosome. Post-translational modification (PTM) of histone tails bookmarks the genome. Together with DNA methylation, they provide an important part of epigenetic memory regarding transcriptional events [11,12]. Epigenetic mechanisms are shown in Figure 1.

While DNA methylation and histone modifications have been studied in detail, the functional consequences of replacing replication-coupled histones with variant histone proteins is less well understood. Histones contribute to DNA template-based regulation of the genome. In mammals, replication-coupled histones are encoded by multiple intron-less gene clusters throughout the genome, synthesized exclusively in the S phase of the cell cycle, and packaged into newly replicated DNA [13]. In contrast, individual genes encode variant histone proteins that are expressed and incorporated into chromatin independent of replication. Each histone variant has a unique temporal expression, and their locus-specific incorporation is mediated by dedicated histone chaperones and chromatin remodelers. Therefore, histone variants are likely to fulfill specific cellular functions that cannot be substituted by replication-coupled histones [14].

Histone variants differ from replication-coupled histones. These differences may range from exchange of single amino acids to the inclusion of additional domains [15]. Thus, histone variants can mediate specific molecular functions in a locus-specific manner. Differences in primary protein sequence may directly impact on physical and bio-chemical properties of the nucleosome and allow for variant-specific PTMs [16]. In addition, histone variants may provide additional binding sites for regulatory factors [13]. The contribution of histone variant-containing nucleosomes in the alteration of chromatin compaction, nucleosome dynamics and transcriptional output can be specific and profound even if only individual amino acids differ between the replication-coupled histone and its variant [17]. Given the ability of histone variants in remodeling the chromatin structure and altering the epigenetic plasticity, recent data have highlighted a role for histone variants in cancer initiation, progression and metastasis [18]. Aberrant expression and mutations of histone variants have been implicated in various cancers [19,20]. While histone variant H3.3 differs from the canonical H3 by only five amino acids, active histone marks such as mono-methylation of histone 3 lysine 4 (H3K4me), acetylation of H3K9 (H3K9ac), mono- or di-methylation of H3K36 (H3K36me1/2) are preferentially enriched on H3.3 compared to canonical H3. These changes increase transcriptional activation of H3.3 enriched loci [20]. Mutations of H3.3 have been detected in neoplasms, including K27M and G34R/V in pediatric gliomas, G34W in giant cell tumor of bone (GCTB) and K36M in chondroblastoma [21,22,23,24,25,26]. Specific histone mutations possess the ability to override or inhibit physiologic gene expression by interfering with different PTMs, histone chaperones and/or chromatin architecture. Recurrent mutations in histones were identified within the histone fold domain by large-scale cancer genome analysis in various cancers (reviewed in [15,27]). Detection of these mutations is used for diagnostic purposes in clinical routine.

## 2. Alteration of H2A Variants in Cancer

Compared to other core histone variant families, the H2A family exhibits the highest sequence divergence, resulting in the largest number of known variants in mammals [28] (Table 1). H2A variants mostly differ in their C-terminal regions and these structural differences result in a multitude of biological functions [29,30], including functional roles in human cancer [31]. H2A.Z has two distinct isoforms, H2A.Z.1 and H2A.Z.2, which are encoded by two non-allelic genes, *H2AFZ* and *H2AFV*, respectively, and driven by independent promoters on distinct chromosomes. Although dysregulation of both H2A.Z isoforms has been linked to various cancers, they exhibit isoform-specific functions [31,32,33,34,35,36]. While H2A.Z.1 plays a pivotal role in liver tumorigenesis, H2A.Z.2 was reported as a driver of malignant melanoma [32,37]. Overall, H2A.Z appears to play a direct role in hormone-dependent breast and prostate cancer [38]. Here, in estrogen receptor α (ERα)-positive breast cancer, H2AZ.1 is incorporated at enhancers of ERα-regulated genes and is required for the recruitment of transcription factor FOXA1, which activate ERα-regulated gene transcription and promote tumor proliferation [39]. Moreover, an integrated approach described a transcriptional regulatory cascade involved in cancer progression and identified H2A.Z to be associated with lymph node metastasis and dismal survival [35,38]. In primary fibroblasts, H2A.Z was identified as a negative regulator of p21 and impairs cellular senescence [40]. Taken together, these findings suggest that H2A.Z may act as an oncogene.

Several decades ago, H2A.X was identified in human cells [41]. Although its histone fold domain shows high similarity to the canonical H2A, the C-terminus of H2A.X was extended and subjected to different PTMs. As the most prominent example, γ-H2A.X refers to phosphorylation at S139, which has been established as a marker for DNA damage [42,43]. Given the critical role of γ-H2A.X in cellular response to double-strand breaks (DSBs), H2A.X was implicated in tumor biology as DSBs may induce genomic instability and gene mutation related to cancer. Mice deficient for both H2A.X and p53 develop lymphomas and solid tumors and therefore show increased susceptibility to cancer [44,45]. H2A.X maps to chromosome 11 and deletions of 11q are frequently detected in hematological malignancies, such as T cell prolymphocytic leukemia and mantle cell lymphoma [46]. Moreover, alterations in H2A.X copy number are described in solid tumors such as head and neck squamous cell carcinoma and breast cancer [47,48].

In contrast to established functions for H2A.Z and H2A.X, the role of H2A.J remains poorly understood. H2A.J is only found in mammals, suggesting a mammal-specific function. H2A.J has been implicated in chronic inflammation and the signaling of senescent cells as its deletion inhibited inflammatory gene expression associated with the senescence-associated secretory phenotype (SASP) in human fibroblasts [49]. Gene expression analyses identified *H2AFJ* (gene encoding H2A.J) as aberrantly expressed in breast cancer [50,51,52], though further functional studies are needed to validate its functional role.

Histone variant H2A.B appears to exert oncohistone features [53]. Expression of H2A.B was reported to shorten S phase, alter splicing and display higher susceptibility to DNA damage, all of which can promote malignant transformation [54,55,56]. Moreover, various cancers show aberrant expression of H2A.B, such as genitourinary cancers and Hodgkin’s lymphoma (HL) [55]. However, molecular targets of H2A.B in cancer remain elusive.

Amongst all H2A variants, macroH2A variants exhibited the most unique structural organization as they harbor a non-histone region at the C-terminus, named the macrodomain, making them the largest known histones. This family of proteins has three isoforms: macroH2A1.1 and macroH2A1.2 are isoforms resulting from the alternative splicing of a mutually exclusive MACROH2A1 exon (previously named H2AFY) (Figure 2A). In contrast, macroH2A2 is encoded by the MACROH2A2 gene (previously H2AFY2). Dysregulation of macroH2A has been implicated in various cancers in a context- and isoform-dependent manner. MacroH2A proteins can promote and stabilize differentiated states and may act as barriers for reprogramming, which has resulted in their perception as tumor-suppressors [13]. However, the two macroH2A1 splice isoforms frequently showed opposing effects in benign and malignant cells [57,58,59,60,61,62,63]. Thus far, it remains elusive whether they have independent roles or neutralize their respective functions. One example of how cancer- and cell type-dependent interactions may influence cellular functions is the case of macroH2A1.1 and its capacity to bind poly ADP-ribosyl polymerase I (PARP-1) [64]. As macroH2A1.1 is an endogenous inhibitor of PARP-1, the relative ratio of both macroH2A1 isoforms abundance affects cellular functions associated to PARP-1 inhibition. For instance, in liver cancer, both tumor suppressive and promoting functions have been described [65,66,67,68]. Nevertheless, recurrent mutations in macroH2A have rarely been described [15]. Specific epigenetic regulators interacting with macroH2A, such as the Polycomb repressive complex 2 (PRC2) component EZH2, are frequently mutated in cancer [69]. Here, it remains to be investigated how these mutations impact on macroH2A-dependent functions. In the following paragraphs, we will discuss the functional domains of macroH2A and the current state of knowledge on the tumor promoting and suppressing functions of macroH2A. We will focus on cancer-associated changes in macroH2A expression, their association with prognosis and review the current evidence on their mechanistic function from cell culture to xenograft models.

## 3. MacroH2A Variants: The Three Functional Domains

MacroH2A proteins are composed of three functional domains that, from N-terminus to C-terminus, are a histone fold, an unstructured linker and a globular macrodomain [13,28]. We will discuss (i) how the histone fold domain, linker and macrodomain mediate molecular and cellular functions, and (ii) the current state of knowledge of isoform-specific findings in cancer development. MacroH2A1 will be used as a collective term for both isoforms and macroH2A for data concerning all three macroH2A variants.

Overall, approximately 62% of the amino acid (aa) sequence of the macroH2A histone fold domain (aa residues 1–123) is identical to its counterpart histone H2A [57,73] (Figure 2B,C). Despite the difference in amino acid sequence, the crystal structure shows that the histone fold of macroH2A1 proteins is highly similar to that of replication-coupled H2A [28]. In contrast, the structure of the L1 loop significantly diverges from the structure of H2A and this may affect interaction with the second H2A–H2B dimer in the nucleosome. As a consequence, macroH2A–H2B preferentially forms heterotypic nucleosomes with increased stability compared to replication-coupled H2A–H2B-containing nucleosomes [28,74]. The docking domain is responsible for the interaction of H2A–H2B dimers with H3–H4 tetramers [8]. The secondary structure of the macroH2A docking domain is identical to H2A despite differences in the amino acid sequence [28], which is highly relevant for the correct deposition of macroH2A into specific chromatin environments [75,76].

The linker domain has been defined as the disordered stretch spanning from amino acid 124 to 161. However, as aa residues 161 to 180 degrade during macrodomain crystallization, they may also possess some degree of disorder, making the current definition of the linker rather restrictive [28]. The domain resembles the C-terminal half of H1 proteins, responsible for the binding and stabilization of internucleosomal DNA [57,77]. This comparison is based on the ratio of order-imposing and disorder-imposing amino acid content, 10% and 74%, respectively, in the linker of macroH2A and 1% and 73% in the C-terminal end of H1 [78]. The ability of the linker domain to protect internucleosomal DNA has been shown by reduced exonuclease digestion [77]. The linker enables low-salt nucleosome oligomerization, stronger DNA compaction and reduction in chromatin accessibility, in particular in the absence of the macrodomain [78]. The linker domain is lysine-rich, thus potentially increasing its sensitivity to lysine-directed proteases. Moreover, the linker may undergo PTMs, such as phosphorylation on serine 138 [79,80]. Most recently, the biological effects of the linker domain have been associated with chromatin compaction, DNA repair pathways and the three-dimensional architecture of heterochromatin [81,82]. Similarity between a part of macroH2A’s linker and the C-terminal tails of histone variant H2A.W that is associated with heterochromatin formation in *Arabidopsis* has been described [83].

The macrodomain of the macroH2A histone is a unique property and is composed of the C-terminal amino acids (aa 161 ff.) (Figure 2C). Out of 12 macrodomain-containing proteins found in humans [84], macroH2A is the only chromatin component. Macrodomains have a characteristic fold of seven β-sheets and five α-helices [28,85], and they have been shown to bind ADP-ribose (ADPR) as single molecule, chains or as PTMs, occasionally with hydrolytic activity [86]. MacroH2A1.1 binds SirT7 and PARP-1 in an ADP-ribosylation-dependent manner [87,88]. PARP-1 acts as a cellular stress sensor and transfers ADP-ribosyl groups from NAD+ to target proteins and growing ADP-ribose chains [89]. Depending on the relative ratio of macroH2A1.1 and PARP-1 and the intensity of stress signals, the interaction can lead to inhibition of PARP-1, thereby inhibiting PARP-1-dependent processes such as DNA repair and reducing nuclear NAD+ consumption [63]. Furthermore, the macrodomains of all macroH2A proteins contribute to the binding of other interaction partners—the PRC2 and histone deacetylases (HDACs) independent of ADP-ribosylation [28,90].

Of the three macroH2As, only macroH2A1.1 is able to bind ADPR moieties [91], whereas the macrodomains of macroH2A1.2 and macroH2A2 show no affinity for this metabolite [82,91]. Structural properties of the macroH2A1.2 and macroH2A2 macrodomains can explain changes in their substrate-binding ability: (i) macroH2A1.2 displays a three amino acid insertion within the adenine-binding site of the macroH2A1.1 macrodomain [91]. Both macroH2A1.2 and macroH2A2 show exchange of glycine in the binding pocket compared to macroH2A1.1. Here, G224 is replaced with larger acidic residues, D227 and E227, respectively [82,91]. Of note, the G224E mutation of macroH2A1.1 is unable to bind ADP-ribose [82,91,92]. (ii) MacroH2A2 differs in its shape and chemical properties and is characterized by two proline insertions that alter the loop, which binds the distal phosphate of ADP-ribose in macroH2A1.1 [82]. Thus far, it remains elusive whether structural changes in macroH2A1.2 and macroH2A2 would be compatible with the binding of putative ligands.

Overall, all three domains of macroH2A contribute to its unique molecular function and need to be considered when dissecting the contribution of macroH2A to normal physiological processes and malignant disease.

## 4. The Ambiguous Role of MacroH2A in Transcriptional Regulation

Loss-of-function studies suggested a role of macroH2A in stabilizing differentiated cellular states [13]. However, how macroH2A proteins affect the regulation of gene transcription is not fully understood. While published data suggest that incorporation of macroH2A into nucleosomes affects the biochemical properties of transcriptional chromatin templates, the mechanistic details remain elusive. One major aspect includes the difficulty to associate local macroH2A occupancy with function. Genome-wide studies indicated that macroH2A proteins are broadly distributed and, depending on the algorithm used, are found enriched in up to 30% of the mapped genome [93,94]. This broad distribution contrasts with the finding that in most cell types, macroH2A represents only 1% of the H2A pool [13]. This would correspond to a maximum of 2% regarding all nucleosomes and assuming macroH2A’s preference for forming heterotypic nucleosomes with a replication-coupled H2A [74]. In conclusion, macroH2A appears to be variably positioned in different cells, leading to low local saturation in population-averaged bulk epigenomic data. Overall, macroH2A occupancy has been associated with repressed regions and in particular with the presence of Polycomb repressive complexes and their associated mark, H3K27me3 [81,93]. These findings were consistent with the early observation that macroH2A is enriched at the inactive X chromosome [95], although global enrichment was lower than initially postulated [96]. Conversely, actively transcribed regions such as the gene bodies of expressed genes appear depleted of macroH2A [97].

In the context of repressed chromatin states such as the inactive X or H3K9me3-marked repeat sequences, macroH2A may represent one of several factors contributing to transcriptional repression [81,98]. However, transcriptomic studies performed after genetic ablation of macroH2A revealed deregulation of transcription in both directions [63,93]. A positive contribution to gene expression was particularly pronounced in response to differentiation and stress signals [99]. Therefore, macroH2A may rather contribute to the robustness of gene expression programs [100]. Two hypotheses of how macroH2A may affect transcription have been described: (i) according to the first (‘direct’) hypothesis, the presence of macroH2A directly affects binding of transcription factors to their respective DNA motifs or the function of other chromatin-modifying enzymes. Such an influence could be either positive (activation) or negative (repression). The presence of macroH2A-containing nucleosomes can inhibit the binding of certain transcription factors such as ATF-2 and NFkB [101,102]. On the other hand, macroH2A may facilitate the recruitment of transcription factors and other chromatin-associated proteins by providing additional binding sites, for instance, macroH2A1.1 recruitment of SirT7 and PARP-1 through its isoform-specific binding pocket [88,103]. In addition, macroH2A1.1-dependent recruitment of PARP-1 to euchromatin promotes the acetylation of H2B at two lysine residues in non-malignant cells. Of note, this mechanism may lead to either activation or suppression of genes [103]. Likewise, macroH2A1.2 contributes to the recruitment of the transcription factor Pbx1 in myogenic cells [104]. Other studies have convincingly shown that macroH2A-containing nucleosomes limit chromatin remodeling through SWI/SNF but not ISWI [102,105]. Critically, high local occupancy of macroH2A would be required according to this hypothesis, which is rather not observed. (ii) According to the second (‘indirect’) hypothesis, macroH2A plays a major role in nuclear organization [81,82,106] suggesting that effects on transcription are ‘indirect’ due to influence on higher-order chromatin structure. The impact of macroH2A on chromatin organization is particularly striking on the level of heterochromatin, where large-scale organization can be observed by electron microscopy [81]. Importantly, the unstructured linker region of macroH2A seems to promote the large-scale compaction of heterochromatin [82]. Only compacted heterochromatin can interact with the Lamin B within the nuclear periphery and this interaction is lost in macroH2A-deficient cells [81,106]. Whether macroH2A mediates compaction through Lamin B binding, or vice versa, remains elusive. Regarding potential limitations, most studies linking macroH2A to three-dimensional chromatin architecture have focused on heterochromatin, whose large-scale compaction can be assessed by microscopic methods. However, given the broad genome-wide distribution of macroH2A, it may also affect 3D chromatin architecture in other regions of the genome. In conclusion, macroH2A can affect transcriptional processes, acting as a transcriptional activator or repressor. However, the underlying mechanisms have not been investigated in detail. Its effects may be mediated directly by affecting the recruitment of regulatory factors or indirectly through affecting three-dimensional chromatin organization.

## 5. MacroH2A as a Tumor Suppressor

Reduced expression of macroH2A-encoding genes was observed in highly proliferating tumors of various origins [82] including melanoma [107]. Prognostically, low expression of macroH2A1.1 and macroH2A2 is associated with poor prognosis in several cancers such as lung, colorectal and breast cancer and astrocytoma [108,109,110,111]. The mechanistic causes for downregulation of macroH2A expression have been investigated in myelodysplastic syndrome (MDS), a bone marrow failure disorder characterized by defects in hematopoietic differentiation that eventually may progress to acute myeloid leukemia (AML) [112]. In MDS, downregulation of macroH2A1.1 may be a consequence of mutations in the splicing factor U2AF1 and consecutive changes in alternative splicing. Additionally, loss of chromosome 5q that harbors the MACROH2A1 gene may lead to loss of MACROH2A1 expression [113,114]. Along these lines, several other cancer-associated factors involved in splicing of macroH2A1 were identified. While Quanking (QKI) and Muscleblind Like Splicing Regulator 1 (MBNL1) favored macroH2A1.1 over macroH2A1.2 [61,79,115], DEAD box helicase 5 (Ddx5) and Ddx17 showed biased splicing towards macroH2A1.2 [59]. Functionally, these splicing factors potentially contribute to cancer cell migration and invasion through modulating the alternative splicing of macroH2A1 [59,61].

Recently, genetic perturbation studies have assessed the functional relevance of reduced macroH2A levels. MacroH2A depletion resulted in more aggressive cancer phenotypes in models of teratoma, breast, bladder, prostate and colon cancer [59,109,116,117,118]. In cutaneous melanoma, macroH2A1- or macroH2A2-depleted cells showed increased proliferation and metastatic ability both in vitro and in xenograft studies [107]. The malignant phenotype of macroH2A loss in melanoma but also other cancers was partially promoted by direct transcriptional upregulation of oncoprotein cyclin-dependent kinase 8 (CDK8). Genetic depletion of CDK8 in macroH2A1-deficient melanoma cells attenuated the proliferative advantage related to loss of macroH2A1. In contrast, high expression of macroH2A1 induced G2/M cell cycle arrest in melanoma and osteosarcoma cells [107,119,120]. In line with these studies, an inverse correlation between macroH2A1 and CDK8 was described in breast cancer. Here, E3 ligase Skp2 was identified to mediate macroH2A1 ubiquitination and degradation, in turn promoting CDK8 gene and protein expression that contributed to cancer progression [121]. Importantly, CDK8 is druggable and small molecule inhibitors have been developed [122]. However, the effector genes of macroH2A seem to be cell and cancer type specific. They include calcium channel-encoding genes, the cell fate determinant and cold-shock protein LIN28B in bladder cancer [117,123] and lymphotoxin beta (LTβ) in prostate cancer [124]. In all of these cases, loss of macroH2A leads to increased expression of tumor-promoting effector genes. Reduced levels of macroH2A increase factors associated with cancer stemness and reduced differentiation capacity [65,116,123]. In colorectal carcinoma, the transcription factor ZEB1 is a key mediator of the Wnt signaling pathway and endows cancer cells with a pro-invasive and stem cell-like phenotype [125]. ZEB1 inhibits cellular senescence through repression of macroH2A1 which may be relevant for disease progression [126].

In part, tumor suppressive functions of macroH2A1.1 can be explained by its capacity to bind and inhibit auto-modified PARP-1 [64]. In lung cancer, macroH2A1.1 diminished cell proliferation in a binding pocket-dependent manner that mirrored the pharmacologic inhibition of PARP-1 [61]. Similarly, ectopic expression of macroH2A1.1 inhibited epithelial-to-mesenchymal transition (EMT) by inhibiting PARP-1-dependent activation of the master regulator Snail [60]. This mechanism may impact on the initial steps of tumor metastasis.

Regarding its repressive function, macroH2A acts in conjunction with other epigenetic regulators. MacroH2A-containing nucleosomes co-precipitate PRC2-containing EZH2 [90] and the genome-wide distribution of macroH2A is associated with H3K27me3, the mark induced by EZH2 [93]. In bladder cancer, macroH2A1 suppressed the stemness-promoting gene Lin28B through reciprocal binding to the repressive histone methyltransferase EZH2 and to the promoter region of the Lin28B gene locus [123]. Likewise, macroH2A1.2 was shown to cooperate with EZH2 to establish and maintain a repressive chromatin mark at the locus of the breast cancer-induced osteoclastogenic factor lysyl oxidase (LOX) [124]. Moreover, HDAC1 and HDAC2 may preferentially interact with macroH2A1-containing nucleosomes [28]. Hereby, they may remove active histone marks and establish a more condensed chromatin structure at the locus of macroH2A1 target genes to repress their transcription in cancer cells [117]. Finally, regarding cancer-induced osteoclastogenesis, heterochromatin protein 1α (HP1α) and histone H1 protein (H1.2) were identified as major downstream effectors for macroH2A1.2-mediated oncogene suppression [69]. A number of recent publications have further suggested a relevant function of macroH2A in DNA repair and genome stability [127,128,129,130,131]. Loss of macroH2A may contribute to the clonal evolution of cancer and affect the response to cytotoxic compounds [132]. In summary, macroH2A can mediate tumor suppressive functions by repressing cancer-promoting genes in a cell type-dependent manner.

## 6. MacroH2A as an Oncohistone

While most studies have provided evidence for tumor suppressive functions of macroH2A, others have highlighted potential tumor-promoting functions. The concept of oncohistone function was recently established for cancer-promoting somatic mutations of core histones and histone variants [133]. The concept may be extended to any dysregulation of histone proteins if they function as epigenetic drivers of cancer. A positive correlation between expression levels of macroH2A and advanced tumor phases has been described in different oncogenic contexts. High expression of macroH2A1 was observed in hepatocellular carcinoma (HCC), triple-negative breast cancer, colon cancer, lymphomas and the uveal subtype of melanoma [66,68,134,135,136,137]. In triple-negative breast cancer, high macroH2A expression was associated with poor prognosis [134]. Transcriptional profiling in murine tumor models and human lymphomas revealed a strong correlation between MACROH2A1 and the ability of MYC to induce tumorigenesis. MYC inactivation appears to result in permanent changes in the ability of MYC to function as a transcription factor. It has been shown that MYC inactivation induces chromatin modifications and this state of chromatin structure has been shown to influence the ability of MYC to bind to specific promoter loci [138,139]. Although the mechanism of these changes in chromatin structure remains to be determined, upon MYC inactivation, a subset of genes including MACROH2A1 was permanently repressed. This permanent change in gene expression was associated with the ability of MYC to bind to specific promoter loci, suggesting a potential function for macroH2A1 in cancer development [136]. Consistently, genetic inactivation of macroH2A1 in uveal melanoma cell lines resulted in significant reduction of cellular proliferation, migration and colony-forming capacity that could be related to metabolic changes [137]. Here, again, functional consequences of macroH2A have been associated with its repressive function on gene expression. MacroH2A1 co-occurs and synergizes with DNA methylation in the repression of bona fide tumor suppressor genes such as p16/CDKN2A, MLH1 and Timp3 [135].

Moreover, the relative abundance of macroH2A1 splice isoforms may also be a determining factor. While macroH2A1.1 is preferentially associated with non-proliferative states such as senescence and terminal differentiation, macroH2A1.2 expression was predominantly detectable in less differentiated cell types [13]. An increased macroH2A1.2/macroH2A1.1 ratio correlated with poor survival, tumor growth and me-tastasis in fat-induced HCC [66]. Overexpression of macroH2A1.2 induced lipid uptake, accumulation of excess triglyceride and upregulation of lipogenic gene expression programs [140]. In breast cancer cells, the tumor-promoting function was associated with macroH2A1.2 while macroH2A1.1 showed the opposite functions [59]. In epidermal growth factor receptor 2 (HER-2)-positive breast cancer, HER-2-induced expression of macroH2A1.2, but not macroH2A1.1, amplified HER-2 activity, suggesting a potential positive feedback loop [141]. In HER-2-negative breast cancer, the ratio of macroH2A1.1 versus total macroH2A1 showed correlation with prognosis [134].

Taken together, pro-tumorigenic functions of macroH2A have been reported in various cellular backgrounds. Cancer-promoting functions of macroH2A may also be influenced by its respective splicing isoforms.

## 7. Conclusions

The replacement of a replication-coupled histone by its histone variants has contributed to the complexity of the epigenetic landscape and highlighted the importance of chromatin-associated protein complexes in physiological tissue homeostasis and malignant transformation. According to the current state of knowledge, macroH2A may act as a tumor suppressor or an oncohistone in a context- and isoform-dependent manner (Table 2). Besides its potential use as a biomarker, its involvement in disease development and progression may open novel therapeutic avenues. Given the apparent functional relevance of macroH2A in cancer biology, much attention has been focused on mechanisms regarding the deposition of macroH2A and its effect on the chromatin landscape. To date, chaperones and remodeling complexes involved in the assembly of macroH2A-containing nucleosomes remain poorly understood. Little is known about the co-occurrence of histone variant-specific PTMs and the cross-talk among the PTMs that leads to complex phenotypic outcomes. Quantitative mass spectrometry-based approaches have become a powerful analytical strategy to qualitatively and quantitatively assess for protein abundance and their PTMs [142] and this type of analysis may provide a better understanding of the dynamics of the endogenous interaction between histone variants and their chaperones or chromatin-associated complexes. Furthermore, epigenetic profiling will help to clarify genome-wide macroH2A-mediated epigenetic modification patterns that may influence the transcriptional landscape in a particular cancer type or cell of origin.

## Figures and Tables

**Figure 1 cancers-13-03003-f001:**
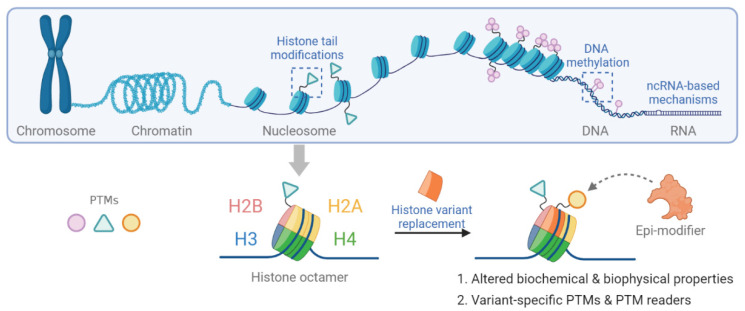
Schematic representation of epigenetic mechanisms including histone tail modification, histone variant replacement, DNA methylation and non-coding RNA-based mechanisms.

**Figure 2 cancers-13-03003-f002:**
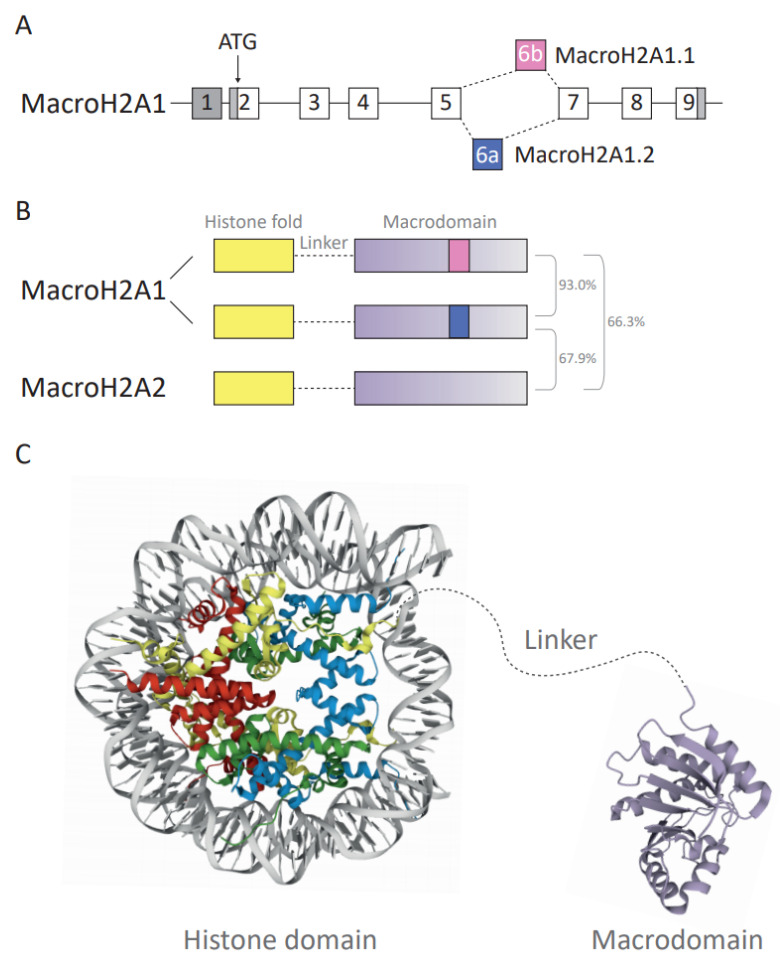
(**A**) Diagram depicting the structure and splicing of the gene encoding macroH2A1 (H2AFY). Gray boxes represent non-coding exons, white boxes represent coding exons. The macroH2A1.1- and macroH2A1.2-specific exons are in pink and blue, respectively. (**B**) Schematic of the three human macroH2A variants’ domain architecture. Total amino acid sequence identity is shown as a percentage. (**C**) Crystallographic structure representation of the macroH2A-containing nucleosome and macrodomains. The protein structure of the unstructured basic linker region depicted in gray is not known. The nucleosome and macrodomain are colored by molecule type. DNA—gray, H2A—yellow, H2B—red, H3—blue, H4—green, macrodomain—purple. The protein structure was generated with protein data bank [70,71] ID: histone fold domain (3REH) [72] and macrodomain of macroH2A1.1 (1YD9) [28].

**Table 1 cancers-13-03003-t001:** Replication-coupled core histones and their variants in mammals.

Canonical Histone	Histone Variants
H2A	H2A.X, H2A.B, H2A.Z.1, H2A.Z.2, H2A.Z.2.2, H2A.J, macroH2A1.1, macroH2A1.2, macroH2A2
H2B	H2BE, H2BW, TH2B
H3	H3.3, H3.Y.1, H3.Y.2, H3.4, H3.5, CENP-A
H4	H4G

**Table 2 cancers-13-03003-t002:** Role of macroH2A in different cancers.

Types of Cancer	Backgrounds	MacroH2A	Potential Mechanisms	References
Anal cancer		MacroH2A2	-	[110]
Bladder cancer		MacroH2A1	Repression of calcium channel genes	[117]
		MacroH2A1.1	Repression of Lin28B	[123]
Breast cancer	Triple-negative	MacroH2A1.1	EMT involvement	[134]
		MacroH2A1.2	Promoting HER-2 signaling	[141]
	Osteoclastogenesis	MacroH2A1.2	Repression of LOX gene	[69]
Colon cancer		MacroH2A1.1	Promoting differentiation	[109]
		MacroH2A1	Repression of p16	[135]
Hepatocellular carcinoma	Steatosis-associated	MacroH2A1	Metabolic implications	[66]
		MacroH2A1	Limiting drug response	[67]
		MacroH2A1	Reducing cancer stemness	[65]
		MacroH2A1	Loss of heterochromatin compaction	[81]
Lung cancer		MacroH2A1.1	Senescence association	[108]
		MacroH2A1.1	Metabolic implications	[61]
Melanoma	Cutaneous	MacroH2A1/2	Regulation of CDK8 genes	[107,119]
	Uveal	MacroH2A1	Potential metabolic implications	[137]
Myelodysplasia	UA2F1 mutation	MacroH2A1	-	[114]
		MacroH2A1.1	Gene regulation of ribosomal biology	[113]
Prostate cancer	Osteoclastogenesis	MacroH2A1	Regulation in LTβ	[124]
Teratoma		MacroH2A1	Promoting differentiation	[116]

Blue indicates macroH2A protein as a tumor suppressor, red indicates macroH2A protein as an oncogene.

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
