# Peer review of "The Role of MacroH2A Histone Variants in Cancer"

_cancers, 2021, doi:10.3390/cancers13123003_

Round 1

Reviewer 1 Report

The authors present a state-of-the-art description of the histone variant macroH2A, from the structure to its role in cancer. Importantly, the review focuses on the epigenetic regulatory role of this H2A variant from a mechanistic point of view. The complexity of this regulation, that as authors say, is highly context- and isoform-dependent, is addressed and clearly described.

Some minor comments:

- Line 55. The composition of the nucleosome as histone homo-dimers leads to confusion. Core histones form heterodimers (as the authors refer later in lines 111-113). I would recommend rephrasing the sentence or remove the homodimer reference.

- Line 127. “[26] [27]” should be “[26, 27]

- Figure 2 legend. The sentence “Total amino acid sequence identity is showed in percentage” lack a period.

- Line 349. If I am not wrong, the reference 121 mention several cancer-associated splicing factors, but not QKI and MBNL1.

- Table 2. This table is not referenced in the text.

Author Response

Response to Reviewer 1 comments:

The authors present a state-of-the-art description of the histone variant macroH2A, from the structure to its role in cancer. Importantly, the review focuses on the epigenetic regulatory role of this H2A variant from a mechanistic point of view. The complexity of this regulation, that as authors say, is highly context- and isoform-dependent, is addressed and clearly described.

We thank reviewer 1 for this positive assessment of our manuscript.

Some minor comments:

Point 1: Line 55. The composition of the nucleosome as histone homo-dimers leads to confusion. Core histones form heterodimers (as the authors refer later in lines 111-113). I would recommend rephrasing the sentence or remove the homodimer reference.

Response 1: We thank reviewer 1 for this important critique. We have adapted the sentence accordingly in the revised version of the manuscript (page 2, text line 55).

Point 2: Line 127. “[26] [27]” should be “[26, 27]

Response 2: We apologize for these formatting errors, which have been corrected in the revised manuscript.

Point 3: Figure 2 legend. The sentence “Total amino acid sequence identity is showed in percentage” lack a period.

Response 3: We apologize for these formatting errors, which have been corrected in the revised manuscript.

Point 4: Line 349. If I am not wrong, the reference 121 mention several cancer-associated splicing factors, but not QKI and MBNL1.

Response 4: We thank reviewer 1 for this correction. This error has been addressed and corrected in the revised version of our manuscript.

Point 5: Table 2. This table is not referenced in the text.

Response 5: We apologize for this mistake. We have referenced table 2 in the text of our revised manuscript.

Reviewer 2 Report

This is a nice comprehensive review written by experts in the field of cancer and histone variants, especially macroH2A.

I only have a few remarks listed by order of appearance in the manuscript:

1. line 53: Should it read chromatin instead of chromosome ?
2. line55: The term replication-coupled should be removed here as it gives the impression that variants won't form a nucleosome.
3. in the paragraph 52-71: I would well add a sentence that the authors are referring here to mammalian histone expression / deposition, as some of the statements are not true for all organisms, e.g. in some species the distinction between replication-coupled and variants is not so evident and e.g. in plants also replicative histones occur as single genes and have introns (e.g. H2A), for other histones e.g. H2B, also variants are devoid of introns.
4.    line 101 maybe revise this part of the sentence : how isoform-specific findings will be distinguished. This is unclear.
5.    line 144: showed should be shown.
6.    Table 1: are the short H2A.B variants the same as H2A.Bbd (is H2A.Bbd old nomenclature?). Please homogenize with the text where only H2A.B is mentioned.
7.    In general e.g. in part #5 the authors only list that this and this variant is reported to be linked to cancer. Maybe it would be interesting to give at least one example for which the mechanism is known and to describe it in more molecular detail.
8.    For the outline of the review it might be consideredto move part 2 (MacroH2A variants: the three functional domains) after part #5.
9.    line 325-36: this sentence is difficult to understand.

Could a summary figure be conceived that illustrates the multiple potential roles of macroH2A in cancer? 

Reviewer 3 Report

Review Cancers: Hsu et al. The role of macroH2A histone variants in cancer.

The authors provide a review on a group of histone variants termed macroH2A. While the recherché is excellent, the organization of the manuscript requires a major revision. 

Major: 
After a short introduction (chapter 1) giving an overview over histone variants, the authors introduce in more detail macroH2A variants (chapter 2). Subsequently, they provide information about cancer associated mutations in histone variants (chapter3), cancer associated alterations in H3 and H4 variants (chapter 4) before they discuss alterations in H2A variants (chapter 5). Only in afterwards (chapter 6) they come back to macroH2A variants. This is not the information expected reading a review about macroH2A variants. In particular, chapter 3 and 4 are out of the focus. I recommend the following: 
Chapter 1 ends with ‘Compared to other core histone variant families, the H2A family exhibits highest sequence divergence resulting in the largest number of known variants in mammals.’
Add the (current) chapter 5 here - maybe split into sub-chapters; H2AX, H2AZ, etc. - and proceed with chapter 6.  Chapter 3 and 4 can be replaced by a reference to review articles focusing on histone variants other than those of H2A; or the authors may discuss their effects in comparison to those of macroH2A at the end of the manuscript. 
In chapter 5, please clarify which alterations are common and which are found in cancer only.
Chapter 7-9: macroH2A in cancer: The authors distinguish between macroH2A as tumor suppressor and onco-histone. So, please, avoid mixing the examples in the related  subchapters as e.g. in chapter 8 (line 361-67).   
Fig. 1 does not provide a clear message. Even after reading the entire manuscript, its statement remains unclear. In particular, it does not clarify why: ‘histone variants are likely to fulfill specific cellular functions that cannot be substituted by replication-coupled histones.’ 

Quantitative mass spectrometry is mentioned for the first time in the Conclusions. Please, explain why it might be helpful. Are their published studies?

Minor:
Please clarify the text: line 416-418. Aberrant MYC is well known to be involved in cancer development; but why does regulation by MYC suggest a functional role in cancer development?
Please avoid multiple discussion of the MacroH2A1.1 binding properties.
Fig. 2 it is not clear why old terms (see line 98, 99) are used.
Abbreviations should be explained by their first use (e.g. H3K27, first use line 181, explanation line 266) 

More details about linker domain PTMs (126-298) would be interesting.

Author Response

Response to Reviewer 3 comments:

The authors provide a review on a group of histone variants termed macroH2A. While the recherché is excellent, the organization of the manuscript requires a major revision. 

Major:
Point 1: After a short introduction (chapter 1) giving an overview over histone variants, the authors introduce in more detail macroH2A variants (chapter 2). Subsequently, they provide information about cancer associated mutations in histone variants (chapter3), cancer associated alterations in H3 and H4 variants (chapter 4) before they discuss alterations in H2A variants (chapter 5). Only in afterwards (chapter 6) they come back to macroH2A variants. This is not the information expected reading a review about macroH2A variants. In particular, chapter 3 and 4 are out of the focus. I recommend the following: Chapter 1 ends with ‘Compared to other core histone variant families, the H2A family exhibits highest sequence divergence resulting in the largest number of known variants in mammals.’ Add the (current) chapter 5 here - maybe split into sub-chapters; H2AX, H2AZ, etc. - and proceed with chapter 6. Chapter 3 and 4 can be replaced by a reference to review articles focusing on histone variants other than those of H2A; or the authors may discuss their effects in comparison to those of macroH2A at the end of the manuscript. 

Response 1: We thank reviewer 3 for these positive and insightful comments on our manuscript. We do agree with the reviewer’s suggestion regarding the order of the chapters and focus of the manuscript. We have shortened the manuscript accordingly and sub-chapters are arranged as recommended by reviewer 3.

Point 2: In chapter 5, please clarify which alterations are common and which are found in cancer only.

Response 2: This statement has been clarified in the revised version of our manuscript (line 121 & line 140).

Point 3: Chapter 7-9: macroH2A in cancer: The authors distinguish between macroH2A as tumor suppressor and onco-histone. So, please, avoid mixing the examples in the related subchapters as e.g. in chapter 8 (line 361-67).

Response 3: We apologize for this confusing terminology. We have corrected the related subchapters accordingly (line 343-344).

Point 4: Fig. 1 does not provide a clear message. Even after reading the entire manuscript, its statement remains unclear. In particular, it does not clarify why: ‘histone variants are likely to fulfill specific cellular functions that cannot be substituted by replication-coupled histones.’

Response 4: We thank reviewer 3 for this assessment. We believe that Figure 1 had been placed in the wrong context. Therefore, we have rearranged Figure 1 and the text accordingly. In the current order, Figure 1 provides schematic representation of epigenetic mechanisms including histone tail modification, histone variant replacement, DNA methylation, and non-coding RNA-based mechanisms and that is supported by the text flow (new text line 60-61).

Point 5: Quantitative mass spectrometry is mentioned for the first time in the Conclusions. Please, explain why it might be helpful. Are their published studies?

Response 5: This is a valid aspect raised by reviewer 3. We have added a paragraph to explain how mass spectrometry approaches may be helpful (page 10, text line 436-446):

“To date, chaperones and remodeling complexes involved in the assembly of macroH2A-containing nucleosomes remain poorly understood. Little is known about the co-occurrence of histone variant-specific PTMs and the cross-talk among each PTMs that leads to complex phenotypic outcome. Quantitative mass spectrometry-based approaches have become a powerful analytical strategy to qualitatively and quantitatively assess for protein abundance and their PTMs [141] and this type of analysis may provide a better understanding of the dynamics of endogenous interaction between histone variants and their chaperones or chromatin-associated complexes. Furthermore, epigenetic profiling will help to clarify genome wide macroH2A-mediated epigenetic modification patterns that may influence the transcriptional landscape in a particular cancer type or cell of origin.”

Included References:

Sidoli, et al., A mass spectrometry-based assay using metabolic labeling to rapidly monitor chromatin accessibility of modified histone proteins. Scientific Reports. 2019. PMID: 31541121.

Kennani, et al., Systematic quantitative analysis of H2A and H2B variants by targeted proteomics. Epigenetics & Chromatin. 2018. PMID: 29329550.

Rea, et al., Quantitative Mass Spectrometry Reveals Changes in Histone H2B Variants as Cells Undergo Inorganic Arsenic-Mediated Cellular Transformation. Mol Cell Proteomics. 2016. PMID: 27169413.

Minor:

Point 6: Please clarify the text: line 416-418. Aberrant MYC is well known to be involved in cancer development; but why does regulation by MYC suggest a functional role in cancer development?

Response 6: Several MYC target genes have relevant roles in different cancers: NAP1l1 has been shown to be a tumor marker for colon cancer, TRIP13 expression was highly elevated in tumor tissues (PMID: 15184677), altered regulation of CCNG1 has been observed in breast cancer (PMID: 10196184), high expression of NOLA2 has been seen in squamous cell lung cancer (PMID: 15889794).

Moreover, incorporation of histone variants can be guided to specific genomic regions by transcription factors, such as MYC. This allows specific and context-dependent gene expression regulation by the variant (in this case, H2A.Z) (Ref: PMID: 12776177, 22081016, 20432434).

For clarification, the following paragraph has been added to the revised version of our manuscript on page 9, lines 396-404:

MYC inactivation appears to result in permanent changes in the ability of MYC to function as a transcription factor. It has been shown that MYC inactivation induce chromatin modifications and this state of chromatin structure has been shown to influence the ability of MYC to bind to specific promoter loci [135, 136]. Although the mechanism of these changes in chromatin structure remains to be determined, upon MYC inactivation, a sub-set of genes including MACROH2A1 was permanently repressed. This permanent changes in gene expression was associated with the ability of MYC to bind to specific promoter loci, suggesting a potential function for macroH2A1 in cancer development.

Point 7: Please avoid multiple discussion of the MacroH2A1.1 binding properties.

Response 7: The discussions have been adapted accordingly.

Point 8: Fig. 2 it is not clear why old terms (see line 98, 99) are used.

Response 8: Terminology has been adapted in the revised version of our manuscript.

Point 9: Abbreviations should be explained by their first use (e.g. H3K27, first use l. 181, explanation l. 266)

Response 9: The abbreviations are explained in the revised version of our manuscript.

Point 10: More details about linker domain PTMs (126-298) would be interesting

Response 10: We agree with review 3. The following paragraph has been added to the revised version of our manuscript on page 5, lines 202-211.

“The ability of the linker domain to protect internucleosomal DNA has been shown by reduced exonuclease digestion [73]. The linker enables low salt nucleosome oligomerization, stronger DNA compaction, reduction of chromatin accessibility in particular in the absence of the macrodomain [74]. The linker domain is lysine-rich, thus potentially increase its sensitivity to lysine-directed proteases. Moreover, the linker may undergo PTMs, such as phosphorylation on Serine 138 [75, 76]. Most recently, the biological effects of the linker domain have been associated with chromatin compaction, DNA repair pathways, and the three-dimensional architecture of heterochromatin [77, 78]. Similarity between a part of macroH2A’s linker and the C-terminal tails of histone variant H2A.W that is associated with heterochromatin formation in Arabidopsis has been described [79].”
